# Calcineurin Gamma Catalytic Subunit PPP3CC Inhibition by miR-200c-3p Affects Apoptosis in Epithelial Ovarian Cancer

**DOI:** 10.3390/genes12091400

**Published:** 2021-09-10

**Authors:** Eleni Anastasiadou, Elena Messina, Tiziana Sanavia, Vittorio Labruna, Simona Ceccarelli, Francesca Megiorni, Giulia Gerini, Paola Pontecorvi, Simona Camero, Giorgia Perniola, Mary Anna Venneri, Pankaj Trivedi, Andrea Lenzi, Cinzia Marchese

**Affiliations:** 1Department of Experimental Medicine, Sapienza University of Rome, 00161 Rome, Italy; elena.messina@uniroma1.it (E.M.); vittoriolabruna96@gmail.com (V.L.); simona.ceccarelli@uniroma1.it (S.C.); francesca.megiorni@uniroma1.it (F.M.); giulia.gerini@uniroma1.it (G.G.); paola.pontecorvi@uniroma1.it (P.P.); maryanna.venneri@uniroma1.it (M.A.V.); pankaj.trivedi@uniroma1.it (P.T.); andrea.lenzi@uniroma1.it (A.L.); cinzia.marchese@uniroma1.it (C.M.); 2Department of Medical Sciences, University of Torino, 10126 Torino, Italy; tiziana.sanavia@unito.it; 3Department of Maternal, Infantile and Urological Sciences, “Sapienza” University of Rome, 00161 Rome, Italy; simona.camero@uniroma1.it; 4Department of Gynecological-Obstetric Sciences and Urological Sciences, Sapienza University of Rome, 00161 Rome, Italy; giorgia.perniola@uniroma1.it

**Keywords:** epithelial ovarian cancer, miRNA, PPP3CC, calcineurin, apoptosis, TCGA and CCLE databases

## Abstract

Epithelial ovarian cancer (EOC) outpaces all the other forms of the female reproductive system malignancies. MicroRNAs have emerged as promising predictive biomarkers to therapeutic treatments as their expression might characterize the tumor stage or grade. In EOC, miR-200c is considered a master regulator of oncogenes or tumor suppressors. To investigate novel miR-200c-3p target genes involved in EOC tumorigenesis, we evaluated the association between this miRNA and the mRNA expression of several potential target genes by RNA-seq data of both 46 EOC cell lines from Cancer Cell line Encyclopedia (CCLE) and 456 EOC patient bio-specimens from The Cancer Genome Atlas (TCGA). Both analyses showed a significant anticorrelation between miR-200c-3p and the protein phosphatase 3 catalytic subunit γ of calcineurin (PPP3CC) levels involved in the apoptosis pathway. Quantitative mRNA expression analysis in patient biopsies confirmed the inverse correlation between miR-200c-3p and PPP3CC levels. In vitro regulation of PPP3CC expression through miR-200c-3p and RNA interference technology led to a concomitant modulation of BCL2- and p-AKT-related pathways, suggesting the tumor suppressive role of PPP3CC in EOC. Our results suggest that inhibition of high expression of miR-200c-3p in EOC might lead to overexpression of the tumor suppressor PPP3CC and subsequent induction of apoptosis in EOC patients.

## 1. Introduction

Epithelial ovarian cancer (EOC) is the second most lethal gynecologic malignancy after cervix cancer worldwide [1]. Therapeutic management of EOC patients is extremely challenging because of the late symptoms and diagnosis when the malignancy reached advanced stages, III–VI, according to the International Federation of Gynecology and Obstetrics (FIGO) classification. More than 90% of the patients are diagnosed with late stage of EOC [2]. Therapeutic treatment spans from surgical debulking of the affected ovaries and immunotherapy, to targeted or chemotherapy or combinatorial therapies [1,3,4,5]. Despite all these treatments, worse prognosis of late diagnosed EOC patients persists and the overall survival (OS) rate is very low [6]. Given the high incidence and mortality of EOC patients, new biomarkers are needed to detect cancer onset at earlier stages, in order to treat and to possibly find a cure for this malignancy.

MicroRNAs (miRNAs) are short single-stranded non-coding RNAs, 21-23 nucleotides, that regulate gene expression by binding to complementary sites at the 3′ untranslated region (UTR) of genes whose dysregulation may lead to cancer, and are involved in different biological processes, such as cell proliferation, programmed cell death (apoptosis), and differentiation [7,8]. In EOC, miRNAs can regulate the expression of tumor suppressors or oncogenes involved in the pathogenesis and progression of this malignancy, thus acting as early diagnostic biomarkers [9]. Several studies showed that miR-200c acts as an oncosuppressor in EOC [3,4], whereas other studies provided evidence of an oncogenic role of miR-200c in these tumors [10]. For instance, high expression levels of miR-200 family were detected in the serum of 163 EOC patients compared to normal samples [11] and they were associated with lymph node metastasis, FIGO stage III-IV, and poor OS. The potential oncogenic or tumor suppressor role of miR-200c in EOC might depend on the histology profile and stage of the ovarian cancer [12], but it could also reasonably depend on the miR-200c target genes, which might be either tumor suppressors or oncogenes, thus having different outcomes in tumor progression.

Calcineurin (CN) is a calcium-dependent serine/threonine phosphatase with a central function in immunity that dephosphorylates different transcription factors, such as the nuclear factor of activated T cells (NFAT), which translocate to the nucleus and regulates the expression of several target genes [13]. This phosphatase is composed by a catalytic subunit (CNA), encoded by the three isoforms (PPP3CA, PPP3CB, and PPP3CC), and a regulatory subunit (CNB), encoded by two isoforms (PPP3R1 and PPP3R2). Other substrates of CN consist of proteins involved in cell cycle and apoptosis [14]. In OC, higher CN levels were positively correlated with OC markers, pathological stage, lymph node metastasis, chemotherapeutic resistance, and reduced OS [15]. Interestingly, the same study showed that CN expression increased significantly in the late stages of Serous carcinoma patients. Among the isoforms encoding for the CN subunits, PPP3CC, also called CNA3, CALNA3, or PP2Bgamma, has been shown to be involved in apoptosis in bladder cancer [16], and reduced expression was significantly correlated to prostate cancer recurrence [17]. However, the role of CNA and, more specifically, PPP3CC has not yet been investigated in EOC.

In the present study, we studied if miR-200c-3p targets PPP3CC in EOC both in vitro and patient derived clinical samples. Using in silico analyses of RNA-seq data from CCLE and TCGA, we confirm a strong inverse correlation between miR-200c-3p and PPP3CC expression in EOC.

## 2. Materials and Methods

### 2.1. Patients and Cell Lines

Laparoscopic ovarian biopsies collected [3,5] prior to surgical intervention and chemotherapy treatment were obtained from 23 EOC patients. All patients were diagnosed with serous ovarian carcinoma grade 3, and, according to FIGO criteria, the tumors were at stage III-IV, indicating that metastasis was occurred. Six out of twenty-three patients had BRCA1 mutation. Normal ovarian tissue biopsies were obtained from 14 patients with benign ovarian cysts and without any previously diagnosed malignancies. All patients involved in this study gave their written consent. The study design was approved by the Ethics Committee of Policlinico Umberto I Hospital, C.E. Ref: 1454/24.07.08, Prot. no. 702/08 (Rome, Italy). The cell line, UWB1.289 + BRCA1, was purchased from the American Type Culture Collection (ATCC). This is a cell line derived from UWB1.289 (ATCC CRL-2945; Manassas, VA, USA), a BRCA1-null human ovarian cancer line, in which wild-type BRCA1 was restored and classified as papillary serous epithelial OC [18]. The cells were grown in 50% RPMI and 50% MEGM (Mammary Epithelial Growth Medium), supplemented with 3% FBS, 2 mM L-glutamine, 100 µg/mL of streptomycin, and 100 U/mL of penicillin. Antibiotic G418 was added twice a week in the cell culture medium at a concentration of 0.2 mg/mL. We chose this cell line because it better represents patients’ cohort characteristics for our study, and expresses high endogenous levels of miR-200c-3p and intermediate to low levels of PPP3CC. It is also a high-grade EOC and has no BRCA1 mutations.

### 2.2. MiR-200c-3p Expression in Public Databases and mRNA in EOC Patients through CCLE and TCGA

Expression of miR-200c-3p was analyzed in publicly available data from CCLE portal (https://portals.broadinstitute.org/ccle/data (accessed on 15 July 2021), considering the normalized expression dataset on miRNA profiling from the Nanostring platform (file CCLE_miRNA_20181103.gct) [19]. The distribution of miR-200c expression was considered at both pan- and OC levels, retrieving the corresponding expression values from 942 and 46 untreated cell lines, respectively. The observed distribution of the expression values allowed us to identify two populations of OC cell lines at high and low miR-200c expression. From these two populations, differentially expressed genes with anti-correlated expression with respect to miR-200c-3p were identified using RNA-seq data retrieved from the CCLE and normalized with the RNA-Seq by Expectation-Maximization (RSEMalgorithm) [20], (fileCCLE_RNAseq_rsem_genes_tpm_20180929.txt). Specifically, a two-tailed unpaired *t*-test was used to match mRNA expression of each gene with high or low levels of miR-200c-3p in the same cell lines. We then explored the differentially expressed genes through TargetScan Human 7.2 (http://www.targetscan.org/ (accessed on 20 July 2021), a bioinformatic tool for miRNA target screening. In addition, we considered mRNA and miRNA data normalized for batch-effects of 477 OC samples (467 primary tumors and 10 solid normal tissues) from TCGA. Specifically, data from gene expression microarrays (level 3) were considered since they include measurements of a higher number of samples with respect to the available (308) TCGA RNA-seq data, which do not provide any measurements for normal tissues. From the 467 primary tumors, 21 samples showing BRCA1 and/or BRCA2 non-silent somatic mutation were filtered out. Then, considering the genes previously selected from the analysis of CCLE and by TragetScan, we performed the following analyses on TCGA data: (1) correlation between the expressions of each gene and miR-200c-3p; and (2) differential gene expression between primary tumors and normal tissues. Finally, enrichment analysis (Fisher’s exact test) was performed on the resulting list of genes using pathway annotations from ConsensusPathDB (http://cpdb.molgen.mpg.de/ (accessed on 23 July 2021) focusing more on genes involved in apoptosis, which were evaluated through literature check in order to narrow down the list of genes to be experimentally validated.

### 2.3. RNA Extraction and RT-qPCR

RNA was extracted from OC patients’ tissues, as previously described [3,4,5]. Briefly, a small piece (5 mm^2^) from each biopsy was homogenized through three cycles of sonication (2 min at 30 Hrtz) in 1 mL of TRIzol™ (Invitrogen; Cat. n. 15596026, Milan, Italy). Extraction was performed following the manufacturer’s instructions. For UWB1.289 + BRCA1 cells, RNA was extracted with the Single Cell RNA Purification Kit (Norgen Biotek Corp.; Cat. n.51800) according to the instructions. MaestroNano micro-scale spectrophotometer (MaestroGen Inc., Hsinchu City, Taiwan) was used to quantitate RNA extracted from each sample and the quality was controlled by electrophoresis in 1% agarose gel. Subsequently, 1μg of RNA of each sample was retrotranscribed with miScript II RT Kit (QIAGEN; Cat. n. 218161 Hilden, Germany), using the HiFlex buffer, which allows amplification of miRNA and mRNA from the same cDNA. Detection of miR-200c-3p, PPP3CC, BCL2, and the housekeeping genes, U6 and GAPDH in each sample, was performed with a miScript SYBR Green PCR Kit (QIAGEN; Cat. n.218073, Hilden, Germany). Detection of miR-200c-3p and U6 was performed using QuantiTect Primer Assays (QIAGEN; Hs_miR-200c_1 miScript Primer Assay, Cat. n. MS00003752; Hs_RNU6-2_11, Cat. n. MS00033740, Hilden, Germany). Whereas, for PPP3CC, BCL2 and GAPDH amplifications, KiCqStart™ (Sigma-Aldrich; primer ID: H_PPP3CC_1; primer ID: H_BCL2L1_1; primer ID: H_GAPDH_1, St. Louis, MO, USA) pre-designed primers were used. All quantitative PCRs were performed through Applied Biosystems\StepOne Software v2.2.2 qPCR machine. The fold change was calculated by the 2^−∆∆Ct^ method normalized to either GAPDH or U6 for PPP3CC and miR-200c-3p, respectively.

### 2.4. Oligonucleotide Transfections

Exponentially growing UWB1.289 + BRCA1 cells were plated in a six-well plate at 0.3 × 10^6^ cells per well, to reach 70–80% confluency. The cells were transfected with a 40-nM MISSION^®^ Synthetic hsa-miR-200c-3p inhibitor or with the corresponding control inhibitor (MISSION^®^ Synthetic microRNA inhibitor, NCSTUD002, Sigma Aldrich St. Louis, MO, USA), using DharmaFECT Duo Transfection Reagent (Dharmacon; Cat. n. T-2010-02, Lafayette, CO, USA), according to the manufacturer’s instructions. Cells were re-transfected at 72 h and were harvested at 144 h post-transfection for further experiments, such as RT-qPCR, western blots, clonogenic and migration assays and flow cytometry. The same experimental procedure was used to transfect the cells with siRNA PPP3CC (Sigma Aldrich; SASI_Hs01_00141251; Cat. n. NM_005605, St. Louis, MO, USA) and the corresponding cells and control were harvested at 144 h post transfection. Cells were processed for further experiments, as described above. The transfections were repeated at least three times.

### 2.5. Western Blot Analysis

For protein detection, 0.5 × 10^6^ transfected cells were lysed in a Radio-Immunoprecipitation Assay (RIPA) lysis buffer, as previously described [21]. Whole lysate preparations for each experimental condition were quantified using the Bradford assay measuring the absorption at 595 nm with Biochrom Libra S22 UV/VIS spectrophotometer (Biochrom, Berlin, DE). Forty μg of protein lysate were loaded in 10% polyacrylamide gels and run for 1 h to 1 h and 30 min at 120V. β-actin (C4) (Santa Cruz; Cat. n. sc-47778; [1:5000] Dallas, Texas, United States) was used as a loading control, as well as AKT (Cell signaling; Cat. n. 9272; [1:500], Danvers, MA, USA). BCL2, p-AKT, and cleaved caspase-3 expression in each sample were detected by Bcl-2 (C-2) (Santa Cruz; Cat. n. sc-7382; [1:100], Dallas, TX, USA), Phospho-Akt (Ser473) (Cell Signaling; Cat. n. 9271; [1:100], Danvers, MA, USA) and cleaved caspase-3 (Asp175) (Cell Signaling; Cat. n. 9661; [1:100], Danvers, MA, USA). All WBs were repeated at least two or three times, and densitometry analysis was performed with ImageJ Software (v. 10.2).

### 2.6. Clonogenic and Migration Assays

For the clonogenic assay, 2 × 10^3^ UWB1.289 + BRCA1 cells from each transfection were plated in triplicates for each experimental condition, in a six-well plate. For the colony formation as well as for migration, the cells transfected with anti-miR-200c-3p or with siRNA-PPP3CC, respectively, were plated at 144 h post-transfection. For the clonogenic assay, the cells were left for 10 days to grow and the medium was replenished twice a week. For the migration assays, at 144 h post-transfection, 50 × 10^3^ transfected OC cells with the correspondent controls were plated in duplicates for each experimental condition into BD Falcon™ Cell Culture Inserts with 8-µm pore polycarbonate filters (Falcon; Cat. n. 353097, Torino, Italy), each containing 3% FBS cell culture medium. Each transwell was then placed in a 24-well plate containing 12% FBS cell culture medium and incubated at 37 °C for 24 h. Cells were fixed in methanol 100% and stained with crystal violet 0.1% in methanol. Transwell were rinsed at least three times with tap water. For each experimental condition, 6 areas were photographed under a light microscope EVOS xl (AMG) at 10× or 20× magnifications. To estimate the differences of cell clonogenicity and migration between the controls and transfected cells we used the count tool of Adobe PhotoshopCC2017. The number of colonies (clonogenic assays) and the number of migrated cells from each selected areas of each well were plotted in histograms to evaluate any differences between the controls and the transfected cells. Both assays were performed in triplicates and twice.

### 2.7. Apoptosis Assay

Transfected cells with anti-miR-200c-3p, siRNA-PPP3CC and their corresponding controls were processed for apoptosis using PE Annexin V Apoptosis Detection Kit I (BD Pharmingen™, BD bioscience Franklin Lakes, New Jersey, United States). Cell culture medium was replenished one day before assaying apoptosis. At 144 h post-transfection, the cells were trypsinized and counted with trypan blue solution (Cat. n. T8154, Sigma Aldrich, St. Louis, MO, USA). A quantity of 1 × 10^5^ cells for each experimental condition was washed in 1 mL of 1× binding buffer and centrifuged at 1000 rpm. The supernatant was carefully discharged, and the pellets were suspended in 100 μL of binding buffer. Subsequently, 5 μL of PE Annexin V and 7-AAD were added in the cells and they were left for 15 min at RT in the dark. In each sample, 400 μL of 1x binding buffer was added and all samples were acquired within 1 h on the CytoFLEX Flow Cytometer (Beckman coulter Life Sciences, Indianapolis, IN, USA). Acquisition analysis was performed with CytExpert software to distinguish and estimate the percentage of living, early, and late apoptotic cells. The experiments were repeated at least three times.

### 2.8. Statistical Analysis

For all pairwise comparisons, a two-tailed unpaired *t*-test was performed to assess for differential expression, while correlation was evaluated by Spearman’s rank correlation coefficient. Analyses and visualization were performed in R language (v. 4.0.3). For multiple comparisons, *p*-values were adjusted using Benjamini-Hochberg false discovery rate. For all statistical tests, (adjusted) *p*-values less than 0.05 were considered statistically significant.

## 3. Results

### 3.1. Inverse Correlation Analysis of miR-200c-3p and mRNAs in OC Cell Lines by Mining CCLE and TCGA Data

We previously showed, by using RNA-seq data from 46 OC cell lines in CCLE database, that miR-200c-3p was inversely correlated with oncogenes, such as c-myc and β-catenin [4]. Here, we performed a much more comprehensive evaluation of all the potential target genes of miR-200c-3p, by exploring mRNA/miRNA expressions in both CCLE and TCGA databases. Specifically, we narrowed down the list of miR-200c-3p gene targets through the following criteria: (1) significant inverse correlation with miR-200c-3p in CCLE and TCGA; (2) algorithm prediction of miRNA/mRNA interactions using TargetScan; (3) differential expression between tumor and normal ovarian tissues using TCGA data; and (4) enrichment analysis on the corresponding pathway annotations and check of the literature, using a PubMed search engine.

We first noticed in CCLE that miR-200c-3p showed a bimodal distribution of its expression both at pan-cancer level and across the 46 OC cell lines available from the database (Appendix A). From this distribution, it was possible to identify two specific groups of OC cell lines showing either low (log2 expression <6, in 24 cell lines) or high (log2 expression >10, in 18 cell lines) miR-200c-3p levels, excluding four cell lines at medium expression. After applying a *t*-test on the gene expression data in these two groups of cell lines, 644 genes were selected as differentially expressed. Among these genes, 230 were found inversed correlated compared to high/low levels of miR-200c-3p expression. Predictions on these 230 genes by TargetScan identified 34 genes with conserved 8mer, 7mer, or 6mer sites matching the seed region of miR-200c-3p. To further narrow down the list of genes, inverse correlation analysis with respect to miR-200c-3p was also performed across 446 TCGA OC sample tissues: 16 of the 34 genes resulted significantly anti-correlated with miR-200c-3p. Finally, by evaluating the differential gene expression between primary tumors and normal ovarian tissues, 11 potential target genes of miR-200c-3p in OC were identified. The obtained results for CCLE and TCGA on these genes are shown in Appendix A.

Furthermore, to find out if there are any common pathways for these 11 genes, we performed enrichment analysis. We found that there were no common pathways among these genes. However, we used the pathway annotations to choose which gene could be more related to oncogenic processes, such as apoptosis. We found that PPP3CC was mainly related to apoptotic pathways (Appendix A). In addition, there are very few studies reporting that PPP3CC functions as an oncosuppressor in castration-resistant prostate cancer (CRPC) [22] and in bladder cancer [16], but there is no information about its role in OC.

Therefore, we focused our study on the role of PPP3CC in EOC in relation to mir-200c-3p. Figure 1A,B show the statistically significant inverse correlation between PPP3CC and miR-200c-3p expression found in CCLE and TCGA data from the analyses described above, respectively. Furthermore, TargetScan algorithm predicted the existence of two miR-200c-3p responsive elements (MREs) on two different sequences at the 3′UTR of PPP3CC (Figure 1C).

### 3.2. MiR-200c-3p Is Inversely Correlated with PPP3CC in EOC Patients

We performed microarray data analysis from TCGA to estimate the fold change expression of PPP3CC in 446 EOC biopsies compared to the normal ovarian biopsies (*n* = 10). There was a small but significant decrease in PPP3CC expression (0.7 fold change), with respect to 10 controls (Figure 2Ai and Appendix A). To assess if PPP3CC shows the same pattern also in 23 EOC patients cohort, we performed RT-qPCR analysis. Results showed that PPP3CC expression was low in these patients, compared to the controls with an average fold change in 0.69 (Figure 2Aii), similar to the one obtained through TCGA (Figure 2Ai). On the other hand, in the same cohort, miR-200c-3p expression was significantly higher, compared to the controls (Figure 2Bi), confirming the inverse correlation with PPP3CC seen in CCLE and TCGA data (Figure 2Bii), showing a Spearman correlation equal to −0.47 (*p* = 0.025). Interestingly, no inverse correlation was found between miR-200c-3p and PPP3CC in the 14 controls (Appendix A).

### 3.3. MiR-200c-3p Downregulation Decreases Proliferation and Migration in EOC Cells

Since miR-200c-3p expression was higher in EOC patients than in the controls, we investigated the function of this miRNA in UWB 1.289 + BRCA1 cell line. We chose UWB 1.289 + BRCA1 cell line because it has higher endogenous levels of miR-200c-3p than its parental counterpart, UWB 1.289 (Appendix A), and represents the three main characteristics of the majority of patients’ cohort: BRCA1wt, increased endogenous levels of miR-200c-3p, and was derived from a patient affected with serous EOC. UWB 1.289 + BRCA1 cells were transiently transfected with a miR-200c-3p inhibitor and its corresponding negative control for 144 h. At this time point, we observed that the inhibition of miR-200c-3p had a major phenotypic impact in this cell line. Expression of miR-200c-3p and PPP3CC were assessed by RT-qPCR. Inhibition of miR-200c-3p (Figure 3Ai) was followed by a PPP3CC increase at both the transcriptional and protein level (Figure 3Aii). Decreased levels of miR-200c-3p coincided with reduced cell proliferation (Figure 3Bi,ii) and migration capacity (Figure 3Ci,ii) of UWB 1.289 + BRCA1 cells. These results demonstrated that high levels of miR-200c-3p in UWB 1.289 + BRCA1 cell line contributed to its pro-proliferative phenotype. Thus, this miRNA seems to act as an onco-miRNA, through inhibition of PPP3CC expression. To further confirm that PPP3CC is a target of miR-200c-3p, the UWB 1.289 cell line was transiently transfected with mimics (Appendix A). Over-expression of miR-200c-3p inhibited PPP3CC expression (Appendix A) and increased the clonogenicity of this cell line (Appendix A).

### 3.4. PPP3CC Knock Down in UWB 1.289 + BRCA1 Cell Line Induced Proliferation and Metastasis

To assess if miR-200c-3p has any role in proliferation and migration through modulation of PPP3CC, the same cell line was transfected with a siRNA against PPP3CC. The cells were transfected for 144 h. At this time point we noticed a decrease in PPP3CC transcript and protein expression, with both being reduced almost half fold compared to the controls (Figure 4Aii). Colony formation assay showed a significant increase in the ability of the cells to form colonies (Figure 4Bi,ii). In addition, there was a significant increase in the migratory capacity of OC cells (Figure 4Ci,ii). MiR-200c-3p expression was not affected after PPP3CC specific K.D. (Figure 4Ai), thus suggesting that miR-200c-3p was able to sustain the proliferation and migratory ability of OC cells. The increased capacity of UWB 1.289 + BRCA1 to form colonies and the ability to migrate after K.D. of PPP3CC supports the hypothesis of its function as a tumor suppressor.

### 3.5. Inhibition of miR-200c-3p and PPP3CC, in EOC Transfected Cells, Has Opposite Effects in Apoptosis and AKT Signaling Pathway in EOC Transfected Cells

Our results showed that miR-200c-3p inhibition released PPP3CC expression which negatively affected the clonogenic ability and migration of UWB1.289 + BRCA1 cells. Contrarily, knockdown of PPP3CC enhanced the proliferative capacity and migration in the same cell line. Thus, we investigated whether inhibition of miR-200c-3p or PPP3CC affects apoptosis in UWB1.289 + BRCA1 cells. We performed the AnnexinV/7AAD apoptosis assay through FACS instrument to estimate the percentage (%) of apoptotic cells after inhibition of miR-200c-3p endogenous levels, comparing to the control (Figure 5A). The sum of the Annexin V % (indicating early apoptotic cells) and double positive AnnexinV/7AAD % (indicating late apoptotic cells) in anti-miR-200c-3p cells showed a total increase in apoptosis (52%) compared to the control (39%) (Figure 5Ai). In contrast, K.D. of PPP3CC decreased the total % of apoptotic cells from 42% in the control to 30% (Figure 5Aii). To investigate the molecular mechanisms by which miR-200c-3p inhibition—and consequently the increase in PPP3CC—in the UWB1.289 + BRCA1 cell line might affect proliferation, migration, and apoptosis, we performed protein analysis of serine/threonine protein kinase AKT, known to be involved in apoptosis, cell proliferation, and cell migration [23]. Inhibition of miR-200c-3p seemed to decrease p-AKT expression, whereas K.D. of PPP3CC increased p-AKT expression, whilst total AKT was not affected (Figure 5Bi,ii, first and second panel). Since cyclin-dependent kinase inhibitor 1 (CDKN1A) or p21 modulates cell proliferation and apoptosis, we checked its expression [24]. We found that p21 expression was increased after miR-200c-3p inhibition and, in contrast, decreased after PPP3CC K.D (Figure 5Bi, ii, third panel). Further, we investigated the expression of Bcl-2 anti-apoptotic gene and the apoptotic effector, caspase-3. In UWB1.289 + BRCA1 cells transfected with anti-miR-200c-3p, Bcl-2 expression was decreased and caspase-3 increased (Figure 5Bi, fourth and fifth panel). Contrarily, there was an increase in Bcl-2 expression and a decrease in caspase-3 (Figure 5Bii, fourth and fifth panel). These results sustain the pro-survival function of miR-200c-3p, through the inhibition of PPP3CC.

### 3.6. Network Analysis and Experimental Validation Reveals a Downstream Regulatory Activity of PPP3CC on Apoptosis

To investigate if PPP3CC regulates Bcl-2, we analyzed its expression after transfecting miR-200c-3p inhibitor and PPP3CC siRNA in UWB.1289 + BRCA1 cells. We showed that Bcl-2 mRNA levels decreased in transfected cells with anti-miR-200c-3p, whilst it increased after K.D. of PPP3CC (Figure 6A). Pathway annotations from ConsensusPathDB showed that PPP3CC gene belong to Reactome pathway “Intrinsic Pathway for Apoptosis” [25]. Figure 6B displays the corresponding pathway network, showing that PPP3CC is essentially involved with a cluster of genes mainly interacting with BCL2-associated agonist of cell death (BAD) and BCL2.

## 4. Discussion

MiR-200c has emerged as an important biomarker and therapeutic target in EOC. Mounting evidence supports the dual role of miR-200c-3p in the progression of OC. On the one hand, restoration of miR-200c-3p in an inducible in vivo model of EOC, decreased tumor burden and enhanced sensitivity to paclitaxel [26]. On the other hand, Cao et al., showed that, in comparison to normal controls, miR-200c overexpression in EOC patients biopsies was associated with aggressive tumor progression and advanced stage, suggesting its role as independent prognostic factor in EOC [3,27,28]. In agreement with the latter study, the data presented here support that high expression of miR-200c-3p in EOC patients might be an indicator of progressive malignancy. Thus, the discovery of novel target genes of this miRNA might help to elucidate different molecular pathways involved in EOC progression.

We performed whole-transcriptome analysis in CCLE and TCGA datasets to identify novel genes whose expression is significantly inverse correlated with miR-200c-3p, based on differential expression between tumor and normal ovarian tissues and the presence of miR-200c-3p complementary sequences at their 3′UTR, predicted through Targetscan. PPP3CC resulted as the most interesting potential target gene of miR-200c-3p to be investigated in EOC. This gene encodes the CNA γ catalytic subunit and its function in OC is not yet known. We experimentally validated the inverse correlation between PPP3CC and miR-200c-3p in a cohort of EOC patients and in the UWB1.289 + BRCA1 cell line which resembles the characteristics of these patients. Inhibition of miR-200c-3p in the UWB1.289 + BRCA1 cell line de-repressed its target gene PPP3CC and, as a consequence, clonogenic and migration capacity of this cell line was reduced. The results might seem in contrast to our previous findings regarding the tumor suppressive function of this miRNA [3,4]. However, in the previous studies, we used SKOV3, an EOC cell line with low endogenous miR-200c-3p. In the SKOV3 context, overexpression of this miRNA leads to a decrease in cell proliferation [4]. In contrast, in the present study, inhibition of miR-200c-3p in EOC cells with high endogenous expression of this miRNA increased apoptosis and decreased clonogenicity through induction of PPP3CC. Furthermore, CCLE analysis showed a bimodal distribution of miR-200c-3p expression not only in 46 EOC cell lines but also at a pan-cancer level. Similar context-dependent functional differences of miRNAs have been noted before, particularly for miR-29b in B-cell chronic lymphocytic leukemia (CLL) [29]. As for the increased expression of miR-200c-3p in the present context, the underlying mechanisms could be several, such as the increased transcription and transport across the nucleus and the improved maturation in the cytoplasm. It could also be due to amplification of chromosome where this miRNA is located, as previously reported for other miRNAs [30].

Our study suggests a tumor suppressor function of PPP3CC, for the following findings: (a) significantly low endogenous levels of PPP3CC were found in patients biopsies comparing to the controls; (b) inhibition of miR-200c-3p in UWB1.289 + BRCA1 increased expression levels of its target gene, PPP3CC, thereby enhancing apoptosis with a negative impact on clonogenic ability and migration; and c) downregulation of PPP3CC in the same cell line was associated with decreased apoptosis accompanied by increased capacity to form colonies and ability to migrate. These findings are in concordance with Jeong et al., whose study showed that K.D. of PPP3CC in prostate cancer cells enhanced colony formation [22]. Similarly, in glioma cell lines, PPP3CC downregulation was associated with tumor progression [31].

CNA is pro-tumorigenic in different types of cancer, including EOC [15,32]. As mentioned above, one of the three catalytic subunits of CNA, PPP3CA, was found to be significantly associated with reduced OS in later-stage ovarian serous cancer [15]. Moreover, overexpression of CNB homologous protein isoform 2 (CHP2) increased cell proliferation and metastasis in an OC cell line [33]. It seems that our results are in contrast with the aforementioned studies: PPP3CC increased expression after miR-200c-3p inhibition in UWB1.289 + BRCA1 cells. This led to a decrease in Bcl-2, p-AKT levels, and induced p21 and caspase-3 expression. Similar results were obtained in esophageal cancer cells, after inhibition of miR-200c, which saw an increase in apoptosis, sensitivity to cisplatin, and PPP2R1B expression, which encodes the β-isoform of the A subunit of another serine/threonine phosphatase PP2A, followed by a reduction of p-AKT [34]. In addition, upregulation of p-AKT levels were reported in miR-200c transfected ovarian cancer stem cells (OCSCs) [35]. As in the UWB1.289 cell line carrying BRCA1wt, the same expression pattern of AKT and p21 was showed by Privat et al. in a breast cancer cell line which carries a functional BRCA1 [36]. Interestingly, CNA and Bcl-2 can form a complex in mitochondrial, nuclear, microsomal, and cytosol fractions leading to apoptosis, or, on the other hand, Bcl-2 can block CNA-induced, calcium-dependent cell death [37,38]. There are some controversial pieces of evidence regarding Bcl-2 expression in EOC. Anderson et al. reported that Bcl-2 expression is decreased with tumor progression [39]. Others showed that higher expression of Bcl-2 was associated with increased resistance to drugs, such as cisplatin, in OC cell lines [40,41] and with higher grade EOC [42,43,44].

In agreement with the intrinsic pathway of apoptosis reported in Reactome, the data in vitro confirmed a regulatory interaction between PPP3CC and Bcl-2. In addition, there is only one study showing a direct binding of CNA to Bcl-2 in human neuroblastoma cells [45]. Evidence suggests that Bcl-2 interacted with serine/threonine protein kinase AKT1, known to be involved in apoptosis, cell proliferation, transcription, cell migration [23], and caspase-3 [46]. Bcl-2 and AKT1 interaction was sustained by evidence in curated pathways from BioCarta, as of September 2018 (https://cgap.nci.nih.gov/Pathways/BioCarta_Pathways (accessed on 25 July 2021)), or Kyoto Encyclopedia of Genes and Genomes (KEGG), as of June 2018, involved in platinum drug resistance, colorectal cancer, small cell lung cancer, and prostate cancer [47,48,49]. Further, AKT1 interaction with caspase-3 was supported by different studies [50,51], whereas both had a strong functional link with p21 supported by a relatively larger amount of evidence based on experimental/biochemical data [52], association with curated KEGG, and NCI-Nature Pathways Interaction Database, as of September 2018 [53].

Future experiments in mice models ought to be carried out to better determine PPP3CC ability and to relent unhindered growth of OC. Furthermore, to strengthen the potentiality of miR-200c-3p as a biomarker for this cancer, we are currently collecting plasma from our cohort of EOC patients to perform miRNA profiling. To the best of our knowledge, the present study is the first to show the association between miR-200c-3p and PPP3CC in EOC. Questions concerning how miR-200c-3p inhibition of PPP3CC regulates key regulators of apoptotic pathways and what causes up-regulation of this miRNA in EOC need further investigation. Moreover, it would be interesting to evaluate the tumor suppressor role of PPP3CC through the NFκB pathway as demonstrated in CRPC [22]. Our data, thus, suggest that the newly identified miR-200c-3p target, PPP3CC, may slow down EOC progression and might become an interesting therapeutic tool for the management of EOC patients.

## Figures and Tables

**Figure 1 genes-12-01400-f001:**
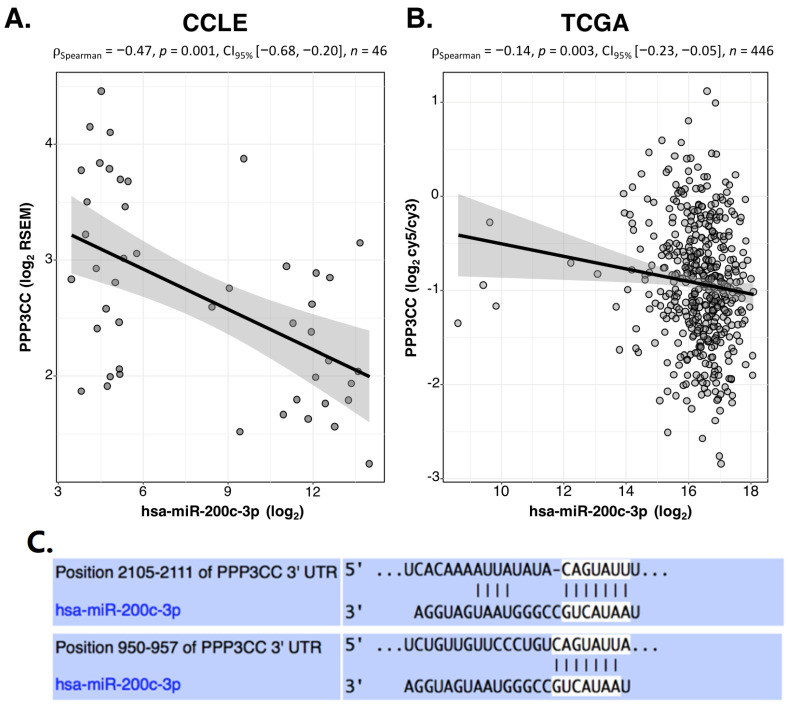
Inverse correlation between miR-200c-3p and PPP3CC through in silico analysis. (**A**). Correlation analysis between miR-200c-3p and PPP3CC in *n* = 46 OC cell lines, based on expression data from CCLE database. Spearman’s rank correlation coefficient was equal to −0.47 with *p*-value 0.001 (**B**). Correlation analysis between miR-200c-3p and PPP3CC in 446 EOC patients, based on expression data from TCGA. Spearman’s rank correlation coefficient was equal to −0.14 with *p*-value 0.003. Data R package ggstatsplot (v. 0.7.2) was used to visualize the correlation analyses. (**C**). TargetScan algorithm was interrogated for miR-200c-3p binding sites to the 3′UTR of PPP3CC. Two binding sites for miR-200c-3p were identified at two different positions of the 3′UTR: at 2105-2111 nucleotides (nt) and at 950–957nt.

**Figure 2 genes-12-01400-f002:**
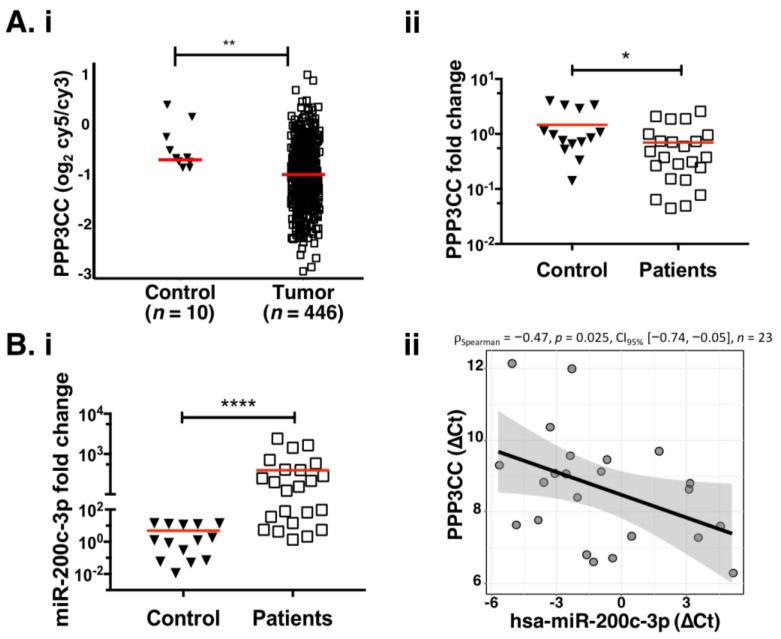
Overall increase in miR-200c-3p and decrease in PPP3CC expression in OC patients. (**Ai**): Expression of PPP3CC in 446 OC sample tissues compared to 10 normal ovarian tissues from TCGA. Log2 lowess normalized ratio of sample signal to human reference RNA signal (Cy5/Cy3) from microarray measurements are reported. (**Aii**): Expression of PPP3CC in 23 patients (white boxes), compared with 14 normal OC biopsies (black arrowheads). The red line represents the mean value of PPP3CC expression in each group. (**Bi**): Expression of miR-200c-3p in the same cohort (controls and patients) as in (**Aii**). (**Bii**): Anticorrelation analysis of miR-200c-3p and PPP3CC in the 23 patients. Spearman’s rank correlation coefficient was equal to −0.47 with *p*-value 0.025. For all the pairwise comparisons, a two-tailed unpaired *t*-test was applied for statistical significance. * *p* < 0.05, ** *p* < 0.01, **** *p* < 0.0001.

**Figure 3 genes-12-01400-f003:**
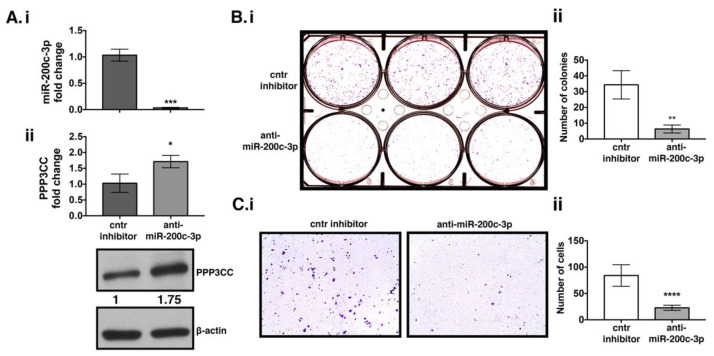
miR-200c-3p inhibition in EOC cells decreases colony forming and migration capacity. UWB 1.289 + BRCA1 cell line was transfected for 144 h with miR-200c-3p inhibitor. (**Ai**): Expression of miR-200c-3p was performed by RT-qPCR. RNU6 was used as a reference gene. (**Aii**): PP3CC expression was assessed through RT-qPCR. GAPDH was used as a housekeeping gene. Fold change expression with respect to the controls was calculated in terms of 2^−^^ΔΔCt^. Western blot (WB) analysis of PPP3CC expression. β-actin was used as a loading control. Values show the fold change in PPP3CC expression in anti-miR-200c-3p transfected cell line comparing to the control (cntr) inhibitor, by Image J densitometric analysis. (**Bi**): 144 h post-transfected cells were placed in a 6-well plate and stained with crystal violet after 10 days to estimate colony formation ability (**Bii**): Graph shows the average number of colonies in each well of the clonogenic assays, calculated with the count tool of PhotoshopCC2017. The experiments were performed in triplicates and were repeated at least twice. (**Ci**): Transwell migration assays at 144 h post-transfection show cells able to traverse the transwell membrane, colored in blue violet. (**Cii**): Graph shows the average number of migrated cells, calculated from each of the 6 selected areas and in each well, by using the count tool of PhotoshopCC2017. Magnification 10×. The white scale bar indicates 400 μm). Two-tailed unpaired *t*-test was applied for statistical significance, * *p* < 0.05, ** *p* < 0.01, *** *p* < 0.001, **** *p* < 0.0001.

**Figure 4 genes-12-01400-f004:**
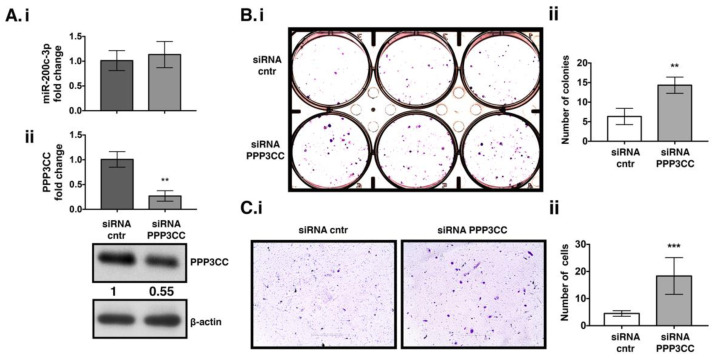
PPP3CC knockdown in EOC cells increases colony formation and migration ability. UWB 1.289 + BRCA1 cells were transfected for 144 h with PPP3CC siRNA (**Ai**): RT-qPCR miR-200c-3p expression, normalized to RNU6 housekeeping gene (**Aii**): PPP3CC transcript was assessed by RT-qPCR. Normalization was performed with the GAPDH housekeeping gene. Fold change expression comparing to the control was calculated as 2^−^^ΔΔCt^. WB analysis of PP3CC expression. β-actin was used as a loading control. Values shows the fold change in PPP3CC expression in UWB1.289 + BRCA1 cells transfected with PPP3CC siRNA comparing to the control (cntr) siRNA, calculated by Image J densitometric analysis (**Bi**): 144 h post-transfected cells were placed in a 6-well plates and stained with crystal violet after 10 days to estimate colony formation (**Bii**): Graph shows the average number of colonies per well, calculated using the count tool (PhotoshopCC2017). The experiments were performed in triplicates and were repeated at least twice. (**Ci**): Transwell migration assay at 144 h post-transfection shows the cells able to traverse the transwell membrane, colored in blue violet. Inhibition of PPP3CC increased the number of migrated cells. (**Cii**): Graph shows the average number of migrated cells, calculated in each of the 6 selected areas in each well, by using the count tool (PhotoshopCC2017). Magnification 10× (white scale bar: 400 μm). A two-tailed unpaired *t*-test was applied for statistical significance, ** *p* < 0.01, *** *p* < 0.001.

**Figure 5 genes-12-01400-f005:**
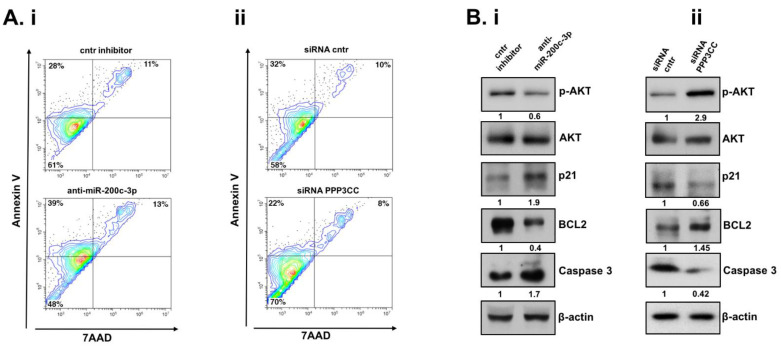
miR-200c-3p inhibition induces apoptosis, whereas reduction of PPP3CC has the opposite effect in EOC transfected cells. (**Ai**,**ii**): Density plots indicate the percentage (%) of cells in each quadrant distributed as: upper left (UL), early apoptotic cells, Annexin V positive. Upper right (UR), late apoptotic cells, Annexin V/7AAD, double positive and low left (LL), living cells, Annexin V/7AAD negative, at 144 h post-transfection. (**Ai**): cntr inhibitor: UL, 28% and UR 11%. Anti-miR-200c-3p: UL, 39% and UR, 13%. (**Aii**): in siRNA cntr: UL, 32% and UR, 10%. siRNA PPP3CC: UL 22%, and UR, 8%. Data shown are from one representative of at least three different experiments. In the density plots the blue, green, orange and red colors represent the distribution of events comparing Annexin V and 7AAD parameters. A gradual increase in event distribution starts from blue color to red color. (**Bi**): UWB 1.289 + BRCA1 cells were transfected with anti-miR-200c-3p and control inhibitor. At 144 h, WB analysis shows p-AKT and AKT, first and second panel, respectively, followed by p21, BCL-2, cleaved caspase-3, and β-actin. AKT and β-actin were used as loading controls. Values indicate the fold change in each protein obtained through densitometric analysis. Each experiment was performed at least three times. (**Bii**): UWB 1.289 + BRCA1 cells transfected with siRNA, PPP3CC, and siRNA control. At 144 h, WB analysis shows p-AKT and AKT, first and second panel, respectively, followed by p21, BCL-2, cleaved caspase-3, and β-actin. AKT and β-actin were used as loading controls. Values indicate the fold change in each protein obtained through densitometric analysis. Each experiment was performed at least three times.

**Figure 6 genes-12-01400-f006:**
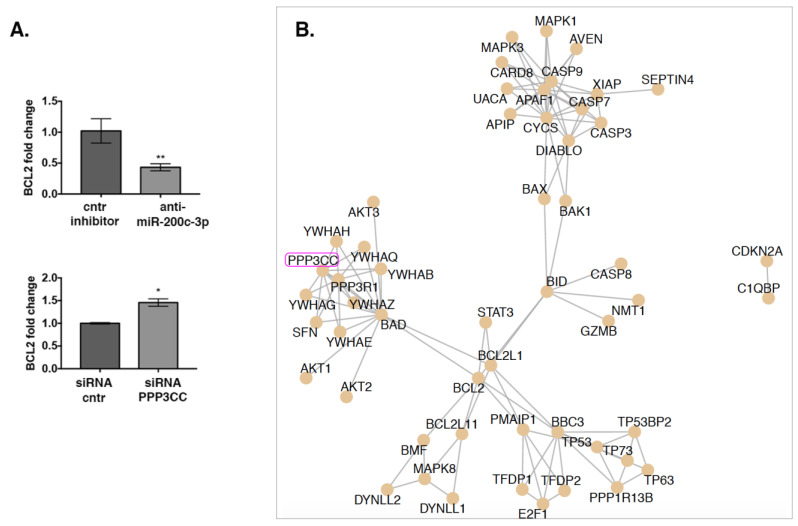
Downstream regulatory activity of PPP3CC on apoptosis. (**A**). Bcl-2 mRNA expression was evaluated through RT-qPCR at 144 h post-transfection. Upper graph: UWB 1.289 + BRCA1 cells were transfected with anti-miR-200c-3p and control inhibitor. Lower graph: UWB 1.289 + BRCA1 cells were transfected with siRNA, PPP3CC, and siRNA control. GAPDH was used as housekeeping gene for PPP3CC expression normalization and RNU6 for miR-200c-3p. Fold change in Bcl-2 was calculated as 2^−^^ΔΔCt^, comparing to the inhibitor or siRNA controls. The experiments were repeated at least three times. Two-tailed unpaired *t*-test was applied for statistical significance, ** *p* < 0.01, * *p* < 0.05. (**B**). Annotation retrieval and visualization of the “Intrinsic Pathway of Apoptosis” from Reactome was performed using ReactomePA R package.

## Data Availability

Data is contained within the article or Appendix A.

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
