# Peer review of "Calcineurin Gamma Catalytic Subunit PPP3CC Inhibition by miR-200c-3p Affects Apoptosis in Epithelial Ovarian Cancer"

_genes, 2021, doi:10.3390/genes12091400_

Round 1

Reviewer 1 Report

1) Line 25, 27 and 29 (anti, expression and interference are one word, please correct)

2) Line 310: Why was 144 hours time point chosen for treatment? please clarify

3) To exclude the possibility that the inhibitor of microRNA used in Figure 3 is indeed operating through inhibition of the microRNA, the authors need to overexpress the miR-200c-3p in the respective cells and demonstrate that PPP3C expression is downregulated with possible greater proliferation of the cells. This experiment will be important to confirm that indeed miR-200c-3p works through PPP3C.

4) Explanation is needed as to why the number of colonies in figure 3Bi (control inhibitor are around 30) versus figure 4Bi (control siRNA is around 5). The number of colonies in control conditions should more or less be the same. Please clarify.

5) Please replace the blot in figure 4ii. The blot is blurry and hard to see.

Author Response

Response to Reviewer 1 Comments:

Question 1, (Q1):Line 25, 27 and 29 (anti, expression and interference are one word, please correct)

Response 1, (R1): We apologize with the reviewerfor the typographic errors. We have now corrected these words, accordingly.

  Q2: Line 310: Why was 144 hours time point chosen for treatment? please clarify

 R2: Thank you for pointing this out. We have performed transfections and proliferation/migration assays at earlier time points, 48 and 72h and 96h. Nevertheless, at these time points there were no marked phenotypic changes. So, we decided to re-transfect the cells at 72h post transfection. Interestingly, at 144h the phenotypic effects of miR-200c-3p inhibition as well as siR-PPP3CC on clonogenicity and migration were more pronounced. According to your suggestion we have clarify this better at Materials and methods section, 2.4, line 161, page 4 and the Results section, 3.3, lines 306-309, page 8. Furthermore, we have corrected the time point as 144h post transfection (line 165, page 4) and we have added the oligo inhibitor and control concentration, 40nM, used in the transfections, line 157, page 4.

 Q3:To exclude the possibility that the inhibitor of microRNA used in Figure 3 is indeed operating through inhibition of the microRNA, the authors need to overexpress the miR-200c-3p in the respective cells and demonstrate that PPP3C expression is downregulated with possible greater proliferation of the cells. This experiment will be important to confirm that indeed miR-200c-3p works through PPP3C.

R3: You have raised a very important point. Initially, we have done experiments of miR-200c-3p over-expression, by transfecting specific mimics, in the parental UWB1.289 cell line. This cell line has lower endogenous levels of miR-200c-3p than UWB1.289+BRCA1. Over-expression of miR-200c-3p had a negative effect on PPP3CC expression and increased clonogenic capacity of this cell line. We have now added a Supplementary Figure 3, the corresponding Supplementary materials and methods and we described these results in Results section 3.3, lines 300-302, 314-317, page 8.

Q4:Explanation is needed as to why the number of colonies in figure 3Bi (control inhibitor are around 30) versus figure 4Bi (control siRNA is around 5). The number of colonies in control conditions should more or less be the same. Please clarify.

R4: Thank you for this important observation. We have repeatedly performed these experiments in different times and in triplicates, as it is shown in Figure 3Bi and 4Bi. We have rigorously maintained the same cell number seeded in six well plates. Nevertheless, we have noticed this difference between the controls. Most likely is due to experimental variation between two different experiments performed in different periods of time.

Q5:Please replace the blot in figure 4ii. The blot is blurry and hard to see.

R5: We apologize for the blurry figure.According to your suggestion, we have improved the resolution of the blot and prepared a new figure 4.

Reviewer 2 Report

In this manuscript Anastasiadou and colleagues were investigating the role of miR200c-3p in regulation of epithelial ovarian cancer development.

Using both in silico and vitro approach authors showed the elevated levels of miR200c3p in ovarian cancer samples. Using the in silico analysis of RNA-seq they found the genes with  inverse expression correlation with miR200c-3p. Authors pointed out the calcineurin (PPP3CC) as a potential clue regulator of apoptosis regulation. Supported by functional studies the calcineurin was proposed to be tan important tumor suppressor, which dysregulation may lead to EOC development.

            This work is very interesting and potentially could be an important contribution to the field, expanding our knowledge on the EOC development mechanism and the role of miR200c-3p in this process.

Specific points:

1.The pictures illustrating the colony formation and proliferation (Fig 3 Bi and Ci as well as Fig 4 Bi and Ci)) are too small and the illustrated phenomenon is not well visible.

  1. The inverted correlation between  miR200c-3p and PPP3CC shown in this paper is clear but the question raises: what causes the miR200 upregulation in EOC cells?  Especially regarding the previous reports, mentioned by Authors in the introduction, showing two opposite roles of miR200c-3p in EOC (from oncosupressor to oncogenic role). Could  authors give comment on that?

  1. As the miR200c-3P level is suggested to be potential tumor biomarker did authors plan to determine its level in RNA isolated from blood plasma?

Author Response

Response to Reviewer 2 Comments:

Question 1, (Q1): The pictures illustrating the colony formation and proliferation (Fig 3 Bi and Ci as well as Fig 4 Bi and Ci)) are too small and the illustrated phenomenon is not well visible.

Response 1, (R1): We apologize for this inconvenience. We have improved the resolution/contrast of both figures and their improved versions are now added.

Q2: The inverted correlation between miR200c-3p and PPP3CC shown in this paper is clear but the question raises: what causes the miR200 upregulation in EOC cells? Especially regarding the previous reports, mentioned by Authors in the introduction, showing two opposite roles of miR200c-3p in EOC (from oncosupressor to oncogenic role). Could  authors give comment on that?

 R2: Thank you for raising this important question. Similar context dependent functional differences of miRNAs have been noted before, particularly for miR-29b in B-cell chronic lymphocytic leukemia (CLL). We have now added the reference PMID: 20936047, Reference n. 29, page 12, line 464.

As for the increased expression of miR-200c-3p in cell line that we have used for this study, the underlying mechanisms could be several, such as, increased miR-200c 3p transcription and transport across the nucleus, its improved maturation in the cytoplasm. It could be also due to amplification of chromosome where this miRNA resides, as has been reported for other miRNAs. We have added a related reference, PMID: 19763153, Reference n. 30, page 12, line 468.

These details are now in the Discussion section, page 12, lines 457-468.

Q3: As the miR200c-3P level is suggested to be potential tumor biomarker did authors plan to determine its level in RNA isolated from blood plasma?

 R3: Thank you for bringing up this important point. Indeed, Oliveira et al, PMID: 31738788, have performed miRNA profiling in presurgical plasma from 95 OC patients. They found miR-200c-3p among the up-regulated miRNAs. Most importantly, the identification of this miRNA in the plasma of OC patients combined with routinely used serum CA-125 levels, may improve the early diagnoses of this lethal malignancy. At present, we are collecting plasma samples from our cohort of EOC patients to study miRNA profiling to strengthen the potentiality of miR-200c-3p as biomarker for this cancer. This has now mentioned in the Discussion section, page 13, lines 516-518.

Round 2

Reviewer 1 Report

Thanks for addressing my comments. Manuscript looks fine as it stands.